# Pragmatics in Language Grounding:
# Phenomena, Tasks, and Modeling Approaches

**Daniel Fried**[1*]    **Nicholas Tomlin**[2*]    **Jennifer Hu**[3]
**Roma Patel**[4,5]    **Aida Nematzadeh**[5]

Carnegie Mellon[1]    UC Berkeley[2]    MIT[3]    Brown University[4]    DeepMind[5]

dfried@cs.cmu.edu    nicholas_tomlin@berkeley.edu
jennhu@mit.edu    {romapatel,nematzadeh}@deepmind.com

## Abstract

People rely heavily on context to enrich meaning beyond what is literally said, enabling concise but effective communication. To interact successfully and naturally with people, LLMs and other user-facing NLP systems will require similar skills in *pragmatics*: relying on various types of context—from shared linguistic goals and conventions, to the visual and embodied world—to use language effectively.

We survey existing grounded settings and pragmatic modeling approaches and analyze how the task goals, environmental contexts, and communicative affordances in each work enrich linguistic meaning. We present recommendations for future grounded task design to naturally elicit pragmatic phenomena, and suggest directions that focus on a broader range of communicative contexts and affordances.

## 1   Using Language in Context

People use language to achieve goals (Wittgenstein, 1953; Austin, 1975; Clark, 1996; Frank and Goodman, 2012), producing effects on other people and the world. To achieve their goals efficiently, people often only sketch their intended meanings: relying on various types of context to allow their conversational partners to enrich meaning beyond what the speaker has literally said. For this reason, language is highly context-dependent; the meaning of even a simple sentence such as "it's nice out today" depends on the situation—it can be an implicit invitation, a statement contrasting the weather with a previous day, or even ironic if the weather is poor. This broad ability to use language in context to achieve goals is known as *pragmatics*.

As large language models (LLMs) and other NLP systems become increasingly integrated into our world, they will require similar pragmatic skills to *use language to interact* successfully and efficiently with people in context. Recent work

has focused on general-purpose models (Tan and Bansal, 2019; Brown et al., 2020; Radford et al., 2021; Bommasani et al., 2021, *inter alia*) that have achieved remarkable performance on a variety of task benchmarks intended to measure literal, semantic meaning. However, current LLMs have little, if any, direct access to most of the types of context that contribute to pragmatic enrichment (see Section 2.1) — they condition on and generate text that is only a noisy observation of the underlying world, interaction context, and communicative effects of language (Andreas, 2022). This lack of direct access to the context and effects of language leads to pragmatic failures. Recent work finds that while LLMs match human performance on some pragmatic tasks, they still struggle with many phenomena, e.g. those that rely on social expectation violations (Hu et al., 2022) and theory of mind (Sap et al., 2022). For multimodal LLMs, contemporary work has found pragmatic failures when reasoning about the intended meaning of entities described in text (Rassin et al., 2022) as well as when reasoning about visio-linguistic compositions of such entities (Thrush et al., 2022).

We believe the time is right to focus on evaluating models on tasks (some existing, but mostly to-be-developed) that require contextual communication abilities: ones that elicit pragmatic phenomena from people, and benefit from pragmatic abilities in systems. These tasks should give a proxy for model performance in grounded interactions with real people, while still facilitating comparing methods and benchmarking progress. In this paper, we center our discussion on the role of grounded and multimodal context in pragmatics, motivated by the rich contexts of typical language use—our understanding of language is shaped by the environments that we use it in (Baldwin, 1995; Bloom, 2002; Tomasello, 2005; Vigliocco et al., 2014; Bender and Koller, 2020; Bisk et al., 2020).

We first survey how various communicative and

---

[*]Equal contribution.

environmental context types elicit pragmatic phenomena. Using these context types and phenomena, we then survey representative grounded tasks and datasets which have been used both to study pragmatic communication in people, and to build goal-oriented multimodal systems. We present tasks along a spectrum of complexity, ranging from constrained reference games to goal-oriented embodied dialogue. We discuss how choices in grounded task design—including environment properties, context types, and communicative affordances—shape the pragmatic phenomena that arise in tasks, and provide suggestions for future task and dataset designers. To model these tasks and phenomena, we give an overview of a range of computational pragmatic approaches that view communication as goal-directed actions by agents in context.

Finally, we suggest further integrations of computational pragmatics with NLP: developing tasks with rich contexts; contextualizing existing tasks; using pragmatic and contextual modeling to allow systems to communicate and interact more successfully and efficiently with people; and tackling challenges of human evaluation and data sparsity.

## 2 Pragmatic Phenomena

In linguistics and cognitive science, *pragmatics* is often defined in contrast to *semantics*. Broadly speaking, semantics characterises the literal meanings of linguistic expressions, whereas pragmatics captures the context-dependent components of meaning, which may contain the bulk of actual communication (Clark, 1997; Casasanto and Lupyan, 2015). Pragmatic communication draws upon many different sources of information, ranging from environmental factors to inferences about other agents' unspoken information and goals. This makes pragmatics both a critical and challenging component for designing NLP systems that interact with people. In this section, we discuss the types of context in which language can be situated and the non-literal inferences that arise as a result of these contextual pressures.

### 2.1 Types of Context

Many aspects of pragmatics involve the maintenance of *common ground*, a set of contextual information shared between communicative partners (e.g., Lewis, 1969; Clark and Brennan, 1991; Traum, 1994; Stalnaker, 2002; Clark, 2015). Key elements of common ground include (1) social and

communicative norms, (2) task goals and collaborative actions, (3) common knowledge, and (4) discourse context. In addition to common ground, we focus on pragmatic reasoning that also requires multimodal context, such as (5) visual information or (6) embodied interaction. See Appendix A for definitions and examples for each of these types of context; we also point readers to Levinson (1983) and Birner (2012) for more discussion.

### 2.2 Roles of Pragmatics

We adopt a use-oriented view of pragmatics (Clark, 1996), highlighting various ways that pragmatic reasoning may be used in grounded language tasks. Our taxonomy is not intended to be fully exhaustive, and we caveat that some categories may partially overlap with one another. For a focus on classic linguistic pragmatic phenomena, like *deixis* and *presupposition*, we refer readers to existing surveys such as Levinson (1983) and Birner (2012).

**Reasoning About Alternatives.** Much of linguistic meaning comes not just from what we say, but from what we do not say. The utterances that speakers choose *not* to say i.e., the set of alternative utterances which are likely in a context, can reveal their intended meanings and mental states (Horn, 1984; Fox and Katzir, 2011; Degen, 2013; Buccola et al., 2022), e.g., *some of the apples are red* likely conveys that some are not, since the speaker did not say all were red. Many of the following roles of pragmatics also often involve reasoning over alternatives.

**Understanding Ambiguity.** Language is frequently ambiguous for many reasons (Piantadosi et al., 2012): ambiguity may be used strategically to achieve communicative efficiency or to remove information that is unhelpful to the task at hand. Moreover, ambiguous instructions often require listeners to reason pragmatically about alternative intentions that speakers might have. For example, when asked to *pass the knife* in a cooking scenario, a pragmatic agent might have to reason about the context to determine whether to provide a butter, bread, or steak knife. By relying on contextual information to resolve ambiguities in situations such as these, pragmatic interlocutors can communicate more efficiently (Solé and Seoane, 2015; Fortuny and Corominas-Murtra, 2013).

**Collaborative Planning.** Many grounded dialogue tasks require agents to coordinate to carry

out joint activities, e.g., collaboratively agree to a goal before executing it. To succeed at tasks like these, participants often must reason about each other's possible goals, for example in a collaborative building setting, inferring that *four planks* can be either a command or a description depending on what effect the speaker is trying to produce on the listener. In environments with many world states, there are a combinatorial number of goals to reason about and actions to take, but a participant can usually only communicate with their partner for a limited time. Therefore, participants must trade-off between communicating efficiently and acting.

**Convention Formation and Abstraction.** Conventions, as characterised by Lewis (1969), are arbitrary but stable solutions to recurring coordination problems that typically form out of the maxims of rational communication (Grice, 1975). For example, a team of workers who communicate with one another daily might initially have lengthy descriptions to refer to certain items, but after a while, might start to develop a common ground of simpler words to refer to them. These abstractions or conventions are hypothesised to emerge as a result of repeated interactions (Garrod and Doherty, 1994). One theory is that conventions form to help resolve ambiguity, yielding more efficient communication at the levels of individuals (Hawkins et al., 2017) or populations (Hawkins et al., 2022).

**Efficiency and Mutual Exclusivity.** For many grounded tasks where the goal is to learn a correspondence between meanings and utterances, pragmatic reasoning can be used to avoid learning degenerate mappings. For example, on learning that a certain label (e.g., *cat*) refers to an object, an agent might use mutual exclusivity to rule out the possibility of another label (e.g., *dog*) also referring to the object (Markman and Wachtel, 1988; Clark, 1988). Models of pragmatic reasoning often induce biases toward mutual exclusivity that can lead to more efficient learning (Wang et al., 2016; McDowell and Goodman, 2019). More broadly, pragmatic reasoning may be used to manage the dual pressures of informativity and conciseness (Zipf, 1949; Horn, 1984; Blutner, 1998), which are explicitly factored into pragmatic models such as RSA (cf. Section 4.2). As a result, pragmatics may lead to communicative efficiency both during language learning and language use.

## 3   Existing Tasks and Environments

In this section, we critically evaluate several well-studied grounded language tasks through the lens of the pragmatic phenomena outlined above. We focus on tasks in multimodal domains that make use of natural language data.[1]

### 3.1   Types of Tasks

Grounded, task-oriented dialogue provides a general setting to study pragmatics. Dialogue tasks provide rich and varied contexts (e.g., different types of common ground, goals, and environments) as well as communicative affordances (e.g., the ability to ask questions, provide information in installments, and adapt to a partner's conventions). These contexts and affordances interact to produce a diverse range of pragmatic behavior (Clark, 1996). However, many of these contexts, affordances, and behaviors are also present in more restricted and controlled tasks for which data collection, analysis, modeling, and evaluation are often more tractable. For example, image captioning tasks simplify data collection and modeling by limiting the number of conversational turns to one; instruction interpretation tasks additionally simplify evaluation (so long as it is possible to carry out and validate actions in the world).

We focus on reference games, image captioning, instruction following, and grounded dialogue tasks that give us a broad characterization of the different properties that tasks might have, as summarized in Table 1.[2] For each task, we specify what type of context is needed, how pragmatic behavior is typically exhibited, and several important elements of task design: partial observability, symmetry, and iterated interaction (see Section 3.2). We present these domains in order of increasing complexity, finding that the most complex grounded dialogue tasks are more likely to involve features like partial observability or symmetry which induce additional pragmatic phenomena.

---

[1]We omit large bodies of work on unimodal pragmatics (Degen, 2013; Jeretic et al., 2020; Choi et al., 2021, *inter alia*) or language that might be grounded, but is synthetically generated (Johnson et al., 2017; Bastings et al., 2018; Zhong et al., 2020).

[2]This is not an exhaustive taxonomy of grounded language learning tasks. For example, VQA (Antol et al., 2015), NLVR2 (Suhr et al., 2019b), the Hateful Memes Challenge (Kirk et al., 2021), and Winoground (Thrush et al., 2022) do not fit perfectly into any of the above categories, although most bear some similarities to image captioning.

| Task (Dataset) | Types of Context | Role of Pragmatics | Par.Ob. | Sym. | Iter. |
|---|---|---|---|---|---|
| **Reference Game** | | | | | |
| *ReferItGame* Kazemzadeh et al. (2014) | Visual | Reasoning about alternatives, understanding ambiguity | ✗ | ✗ | ✗ |
| *Colors in Context* Monroe et al. (2017) | Visual | Reasoning about alternatives, understanding ambiguity | ✗ | ✗ | ✓ |
| **Image Captioning** | | | | | |
| *Abstract Scenes* Andreas and Klein (2016) | Visual, common knowledge | Reasoning about alternatives | ✗ | ✗ | ✗ |
| *Conceptual Captions* Sharma et al. (2018); Alikhani et al. (2020) | Visual, common knowledge, joint goals, norms of interaction | Efficiency considerations | ✗ | ✗ | ✗ |
| **Instruction Following** | | | | | |
| *SHRDLURN* Wang et al. (2016) | Visual | Mutual exclusivity, convention formation, efficiency considerations | ✗ | ✗ | ✓ |
| *CerealBar* Suhr et al. (2019a) | Visual, embodied | Collaborative planning, understanding abstractions and conventions | ✓ | ✗ | ✓ |
| *Hexagons* Lachmy et al. (2022) | Visual, norms of interaction | Understanding abstractions and ambiguity, efficiency considerations | ✗ | ✗ | ✓ |
| **Grounded Dialogue** | | | | | |
| *Cards Corpus* Potts (2012) | Visual, embodied, joint goals, norms of interaction, discourse | Collaborative planning, understanding ambiguity, efficiency | ✓ | ✓ | ✗ |
| *OneCommon* Udagawa and Aizawa (2019) | Visual, joint goals, norms of interaction, discourse | Collaborative planning, understanding ambiguity, efficiency | ✓ | ✓ | ✗ |
| *PhotoBook* Haber et al. (2019) | Visual, common knowledge, joint goals, norms of interaction, discourse | Convention formation, understanding ambiguity, efficiency | ✓ | ✓ | ✓ |

Table 1: Example grounded language learning datasets that involve pragmatic reasoning, organized by task type. The task attributes refer to: partially observable, symmetric, and iterated (multi-turn) interactions. We observe that grounded dialogue and instruction following tasks often involve a broader range of pragmatic reasoning behaviors.

**Reference Games.** Reference games typically involve two players, a listener and a speaker agent. Both players are presented with a shared set of referents, e.g., images, objects, or abstract illustrations, and the speaker is tasked with describing a target referent to the listener, who must then guess the target (Clark and Wilkes-Gibbs, 1986; Gorniak and Roy, 2004; Steels and Belpaeme, 2005; Golland et al., 2010; Frank and Goodman, 2012; Kennington and Schlangen, 2015). An example reference game is the Colors in Context (Monroe et al., 2017) task, in which players are presented with three color swatches and asked to describe one of them. Even simple phrases like *plain blue* may have different meanings depending on visual context in this task.

**Image Captioning.** A broad class of image captioning tasks require producing text to describe an image (Barnard et al., 2003; Farhadi et al., 2010; Mitchell et al., 2012; Kulkarni et al., 2013). Most captioning work has only been implicitly goal-oriented: corpora have been constructed by asking annotators to determine and describe the important parts of an image (Hodosh et al., 2013; Young et al., 2014; Chen et al., 2015). Systems are evaluated on how closely their descriptions match these human-written references, which poses challenges given considerable variation in what annotators chose to describe and how they wrote the descriptions (Anderson et al., 2016).

Other work, particularly in the computational pragmatics literature, has formulated captioning as a contrastive task (Andreas and Klein, 2016; Vedantam et al., 2017; Cohn-Gordon et al., 2018), where a target image must be described to contrast it from other similar, *distractor* images. This setting can be viewed as a scaled-up reference game involving complex visual inputs, and many such pragmatically-motivated variations on standard image captioning have appeared in recent years: Nie et al. (2020) define *issue-sensitive image captioning*, in which models implicitly caption several target images at a time, while Alikhani et al. (2020) train *coherence-aware* captioning models which may vary in the degree of subjectivity or the extent to which inferences about target images are made.

Of the task categories we discuss, image captioning has the most immediate real-world applicability, especially for accessibility e.g., to provide descriptions that could substitute for images

for visually-impaired users on the web (Pont-Tuset et al., 2020). Additionally, practical considerations in this domain often require pragmatic reasoning e.g., specifically describing salient characteristics of an image (e.g., *a man* versus *Barack Obama*), being concise, or describing the relevance of the image to document context. We refer the reader to MacLeod et al. (2017) and Kreiss et al. (2021) for further information on this topic.

**Instruction Following.** Instruction following tasks require a listener to take instructions from a speaker, predicting *trajectories* in an environment (Branavan et al., 2009; Vogel and Jurafsky, 2010; Chen and Mooney, 2011; Tellex et al., 2011; Anderson et al., 2018). Trajectories can be grammar-based actions (e.g., ADD(LEFTMOST(WITH(BROWN)), ORANGE), to specify *add an orange block to the left-most brown block* in the block-stacking setting of Wang et al. 2016), sequences of discrete movements (e.g., between nodes in a navigation graph in Chen et al. 2019; Ku et al. 2020), or continuous sequences (e.g., of orientations in Ku et al. 2020).

A speaker must describe a target trajectory in a way that allows the listener to correctly carry it out in the presence of (often exponentially many) alternative trajectories (e.g., *left* versus *sharp left*). These environments often involve visually-grounded observations (Anderson et al., 2018; Chen et al., 2019; Ku et al., 2020), action hierarchies (Shridhar et al., 2020) or programmatic abstractions (Lachmy et al., 2022) and some parts of the environment may be unobserved to the speaker, the listener, or both (see Section 3.2), causing language to be more ambiguous and context-dependent.

**Grounded Goal-Oriented Dialogue.** We focus on grounded dialogue tasks that involve two-way communication between partners to achieve a shared goal (e.g., Chai et al., 2004; Rieser and Lemon, 2008; Das et al., 2017; De Vries et al., 2017; Kim et al., 2019; Narayan-Chen et al., 2019; Ilinykh et al., 2019).[3] These tasks generalize the one-way communication settings above; however, two-way communication provides additional affordances—allowing players to ask clarification questions, acknowledge understanding, and coordinate actions. For example, in the Cards task (Potts,

---

[3]Our focus is on task-oriented dialogue, given that communicative goals are less explicit in chit-chat settings (but see Kim et al. (2020) for a recent pragmatic treatment).

2012), players collaboratively collect a set of cards in a grid world environment by communicating with other players while moving around to pick up cards. Observability is limited to parts of the environment close to the players, requiring them to pool information, and they must collaboratively plan to agree on one of the multiple possible sets of cards they can collect.

The multi-turn nature of dialogue also necessitates reasoning about past actions and interactions (perform *inference*) and likely outcomes in the future (*planning*). These are particularly evidenced in collaborative reference tasks such as OneCommon (Udagawa and Aizawa, 2019), where players must infer which items they share with their partners, aggregating information over the course of a dialogue. Finally, repeated interactions in dialogue can allow linguistic adaptation. For example, in PhotoBook task (Haber et al., 2019)—a collaborative reference task where players have repeated conversations about photographs—players adapt their language over time to match each other, becoming more efficient over time (e.g., reducing *the strange bike with three wheels* to *strange bike*).

## 3.2 Elements of Task Design

We now outline three especially pragmatically-relevant dimensions to consider when designing tasks and describe how they induce various types of pragmatic phenomena.

**Observability.** In *partially observable* tasks, participants can only see a limited portion of the environment, for example seeing only the parts of the grid closest to them in the Cards task (Potts, 2012). This can make language more context-dependent, in particular creating a dependence on *when* or *where* the language was produced. The most complex partially-observable settings, including all of the collaborative dialogue tasks in Table 1, involve participants observing different views of the environment — requiring them to collaboratively plan to pool their information. Different views can also lead to false agreements where participants believe they have coordinated but actually disagree (Chai et al., 2014; Udagawa and Aizawa, 2019), requiring more explicit pragmatic modeling of the partner's perspective to avoid and resolve ambiguity.

**Symmetry.** Tasks differ in the types of roles performed by the communicating agents, which in turn shapes the type of language produced and

actions taken. We distinguish between *asymmetric* and *symmetric* roles. In an asymmetric setting — e.g., speaker and listener, or teacher and follower — pragmatics may be helpful for production and comprehension of language utterances. Symmetric settings (Vogel et al., 2013a,b) may be more naturalistic and are often used in coordination tasks, although designing such settings is often more complicated. Asymmetric settings (Monroe et al., 2017; Andreas and Klein, 2016) are often the simplest way to introduce pragmatic phenomena, since asymmetry occurs when one agent is missing information.

**Interaction.** The nature of interaction(s) between communicating agents affects the language that is produced. In a *one-turn* interaction, all usable information must be expressed in a single utterance, forcing speakers to balance informativity and conciseness. In *iterated one-sided* interactions, the speaker has the opportunity to respond to the listener's actions before planning each new utterance. Finally, in *dialogue*, agents can freely coordinate and participate in speech acts—they can jointly build common ground, ask clarification questions, and share useful information. These repeated interactions between agents require attention to conversation history, and may give rise to the formation of conventions (e.g., Hawkins et al., 2017).

### 3.3 Evaluating Pragmatic Models

The ultimate goal for user-facing, situated agents is to communicate (1) successfully and (2) efficiently with people. Human evaluations, where agents are paired with people at test-time, are an ideal way to measure this (Walker et al., 1997; Koller et al., 2010; Parent and Eskenazi, 2010; Suhr et al., 2019a, *inter alia*), but are not always feasible to carry out since they complicate controlling and replicating experimental setups. Thus, evaluation often resorts either to static, human-produced corpora or automated model-based evaluations.

**Task success.** Interpretation tasks are typically amenable to corpora-based evaluation. For example, listener agents in reference games can be easily evaluated based on the accuracy of referent selection. In contrast, evaluating language generation tasks for speaker agents is more challenging, given that many classical reference-based automated NLG metrics are unable to measure whether or not generated language will be understood correctly by human listeners (Krahmer and

Theune, 2010; Fried et al., 2018a; Zhao et al., 2021; Gehrmann et al., 2022). Automated proxies for human listeners are models of how people interpret and respond to a system's language, known as *user simulation* or *self-play* (Georgila et al., 2006; Rieser and Lemon, 2011; Lewis et al., 2017; Kim et al., 2019) in dialogue and *communication-based evaluation* (Newman et al., 2020) in reference games, where speaker generations are fed to a listener model and evaluated on task success. Automated models can only give rough indicators of how humans might interpret the system's language. For this reason, we stress the importance of making the evaluation model dissimilar from the system and using human evaluations whenever possible.

**Communicative efficiency.** Beyond task success, a secondary criterion for situated agents is efficient communication. For example, if the language generated by a speaker, although correct, is difficult to understand, this calls for unnecessary interpretation effort from the other agent. To measure whether pragmatic agents enable efficient communication, evaluations can use metrics of communicative cost (Walker et al., 1997) such as time to task completion, utterance length and complexity (Effenberger et al., 2021), measures such as lexical entrainment (Clark and Wilkes-Gibbs, 1986; Parent and Eskenazi, 2010; Hawkins et al., 2020), and quality ratings (Kojima et al., 2021).

## 4 Modeling Pragmatics

In this section, we discuss frameworks that have been proposed to characterize *how* listeners can derive pragmatic meaning, providing a starting point for modeling the phenomena and tasks above.

### 4.1 Gricean Maxims

In his seminal proposal, Grice (1975) argues that speakers and listeners are guided by an underlying *cooperative principle*: taking action to jointly achieve communicative goals, and assuming that other agents are acting similarly. Grice divides this principle up into a set of maxims. However, attempts to directly implement the Gricean maxims computationally (e.g., Hirschberg, 1985a) have had to grapple with substantial underspecification and overlap in Grice's proposal. Later *neo-Gricean* work in linguistics has streamlined the maxims considerably (Horn, 1984; Levinson, 2000) and characterizes many pragmatic effects in terms of the trade-off between speaker and listener effort in achieving

cooperative goals. These approaches have had few direct computational implementations; however, a line of computational work, which we outline in Sections 4.2 and 4.3, *derives* maxim-like behavior through multi-agent modeling rather than by prescriptively implementing the maxims.

## 4.2 Multi-Agent Reasoning

A number of computational frameworks view utterance generation and interpretation using a multi-agent or game-theoretic lens (Rosenberg and Cohen, 1964; Cohen and Levesque, 1990; Golland et al., 2010; Jäger, 2012; Franke, 2013). We focus on one representative of these, the Rational Speech Acts (RSA) framework (Frank and Goodman, 2012; Goodman and Frank, 2016), as it has been successfully applied across a range of grounded language settings.

RSA defines a recursive reasoning process where speakers and listeners model each other's goals and interpretations. A *rational speaker* chooses utterances using an embedded model of how the listener will likely interpret utterances. A *rational listener*, in turn, reasons counterfactually about a rational speaker generating language in this way—reasoning about why the speaker choose an observed utterance rather than alternatives—which can resolve ambiguity in the speaker's utterances.

A variety of work has also applied RSA to improve performance of NLP systems on a range of tasks involving complex natural language utterances, including reference games (Monroe et al., 2017), instruction following and generation (Fried et al., 2018a,b), image captioning (Andreas and Klein, 2016; Cohn-Gordon et al., 2018), summarization (Shen et al., 2019), MT (Cohn-Gordon and Goodman, 2019), and dialogue (Kim et al., 2020; Fried et al., 2021). A number of rational communication frameworks also include noteworthy variations on the core RSA setup, include varying the utility function (Zaslavsky et al., 2021), modeling mis-aligned objectives (Asher and Lascarides, 2013), using deeper levels of recursive reasoning between agents (Wang et al., 2020), and non-linguistic communication (Hadfield-Menell et al., 2017; Jeon et al., 2020; Pu et al., 2020).

One key limitation of RSA is that it models speakers as choosing their utterances from a known and fixed set of candidate utterances. A second notable limitation is that, with a few exceptions (e.g., Khani et al., 2018), applications of full recursive reasoning frameworks have been limited to single-turn interactions. However, the multi-turn approaches that we outline in Section 4.3 allow modeling repeated interactions by making the framework simpler along certain axes (e.g., removing higher-order theory-of-mind).

## 4.3 Multi-Turn Approaches

A variety of approaches to multi-turn pragmatics have arisen in work on task-oriented dialogue. Many of these treat communication as goal-directed decision-making under uncertainty (Rieser and Lemon, 2011; Young et al., 2013), and can be broadly viewed as generalizing the single-turn frameworks of Section 4.2. For generation, a variety of dialogue systems explicitly *plan* utterances or speech acts to convey information to their partners (Cohen and Perrault, 1979; Traum, 1994; Walker et al., 2004; Rieser and Lemon, 2009; Kim et al., 2020, *inter alia*). For interpretation, many systems *infer* the latent intent or state of the user (Allen and Perrault, 1980; Paek and Horvitz, 2000; Williams and Young, 2007; Schlangen et al., 2009; Young et al., 2013, *inter alia*).

Planning and inference are classic AI tasks with broad applicability, and most of the works above are closely related to general machinery developed for *decentralized POMDPs* (Bernstein et al., 2002; Oliehoek and Amato, 2016). However, given computational challenges, past work on algorithmic applications of POMDP algorithms to communication have focused on domain-specific formalisms (the works above) or restricted language settings (Zettlemoyer et al., 2008; Vogel et al., 2013a; Hadfield-Menell et al., 2016; Foerster et al., 2019; Jaques et al., 2019). To enable pragmatic modeling and interaction with people in naturalistic grounded dialogue settings, future work might draw on further progress that the multi-agent reinforcement learning and planning communities make on these underlying algorithmic challenges.

## 5 Discussion

### 5.1 Building Pragmatically-Informed Tasks

Pragmatics becomes most essential — and most challenging — in grounded and interactive settings, which have the richest contexts of language use. When people can rely on shared dialogue, environmental, and task contexts to convey and decode meanings, pragmatic phenomena emerge that have so far been understudied in NLP, such as conven-

tion formation (Section 2.2). While pretraining on multimodal (Lu et al., 2019; Sun et al., 2019; Tan and Bansal, 2019; Radford et al., 2021) and interactive (Stiennon et al., 2020; Shuster et al., 2022; Bai et al., 2022) data has driven recent progress in their respective domains, training data and evaluation setups for settings that are both grounded and interactive remain limited in scope, size, and ecological validity (De Vries et al., 2020). This sparsity poses both a challenge and an opportunity for pragmatic language use.

We encourage future work to focus on realistic interactive scenarios which contain a wide range of pragmatic phenomena. As shown in Table 1, grounded, multi-turn dialogue tasks with partial observability and symmetry often encompass the widest variety of pragmatic language behavior. To date, relatively few tasks have all of these properties, although there are a few notable exceptions (e.g., Potts, 2012; Udagawa and Aizawa, 2019). We argue that such tasks provide useful testbeds for building models of convention formation and collaborative planning, which are currently understudied in the computational pragmatics literature.

In tandem, we also encourage future work to *contextualize* existing NLP tasks in order to bring them closer to real-world applicability. For example, as discussed in Section 3.1, MacLeod et al. (2017) and Kreiss et al. (2021) argue that current image captioning systems fail to meet the needs of visually-impaired users because they do not provide descriptions in the context of the article or of the user's goals. Incorporating and modeling additional context—such as user intent in captioning (Alikhani et al., 2020), interaction history in instruction following (Kojima et al., 2021; Lin et al., 2022), or emotion (Kim et al., 2021) and personality (Wang et al., 2019) in dialogue—may help close the gap between current NLP benchmarks and useful real-world systems.

## 5.2 Pragmatic Modeling and LLMs

We predict that focusing on tasks that require pragmatic capabilities will advance the frontier of NLP, and vice versa. Many of the roles of pragmatics that we outline in Section 2.2 are linguistic manifestations of classical problems in machine learning and AI: inference and model calibration underlie reasoning about alternatives and ambiguity; multi-agent search underlies collaborative planning; adaptation and abstraction learning under-

lie convention formation. Can these problems be solved by (multimodal-)LLMs that condition on the right contexts and are trained on sufficient data, or does there remain a role for explicit pragmatic modeling? What is the best path towards producing human-like pragmatic communicative behavior?

While we view these questions as currently unresolved, we predict that to answer them and to advance grounded NLP, it will prove useful to explore pragmatic modeling and contextually conditioned, multimodal LLMs in tandem. Reasoning about how context enriches meaning (Section 2.1) allows people to say less while still conveying their intentions, and to learn from what could have been said but was not. Similarly, practical benefits of explicit pragmatic modeling include more efficient and accurate communication (Monroe et al., 2017; Khani et al., 2018; Fried et al., 2021) and learning in low data regimes (McDowell and Goodman, 2019; Wang et al., 2016), such as interactive and personalized settings.

LLMs have already begun to be put to use to tackle the grounded pragmatic tasks we outline in Table 1 and Section 3. For example, Kojima et al. (2021) fine-tune a pre-trained GPT-2 model to adapt to people for instruction generation in CerealBar. The pragmatic methods in Section 4 are also compatible with LLMs, e.g., Liu et al. (2023) combine RSA with meta-learning to apply GPT models in an image reference game setting; FAIR et al. (2022) use a large BART model (Lewis et al., 2020) in conjunction with a multi-agent planning procedure in the grounded dialogue game of Diplomacy. As grounded LLM adapters (Alayrac et al., 2022; Merullo et al., 2023; Eichenberg et al., 2022; Koh et al., 2023) continue to improve, we expect to see more work applying LLMs as components of pragmatic models for these grounding tasks.

Regardless of the approaches used, we encourage future work in NLP to evaluate models not just on task success, but also other pragmatic aims: communicative efficiency, understanding of nonliteral language, ability to form conventions (Section 3.3) and consider all approaches that are effective in achieving these goals (e.g., Section 4.2).

## 5.3 Challenges and Open Questions

Going forward, accurate evaluation of pragmatics is one key challenge in grounded language learning. Some recent benchmarks have been proposed to evaluate pragmatic language behavior in large lan-

guage models (Sap et al., 2022; Hu et al., 2022; Ruis et al., 2022), but evaluation in interactive, multimodal scenarios often requires humans or human proxies. As discussed in Section 3.3, some models may be evaluated in self-play, and future work might draw on improved proxies for human interaction from multi-agent reinforcement learning (Strouse et al., 2021). However, human evaluation remains the gold standard in most settings, and we encourage future work to (1) improve human evaluations, taking lessons from the human computer interaction community and (2) focus on tasks that are useful (Bigham et al., 2010; Stiennon et al., 2020), fun (Wang et al., 2017; FAIR et al., 2022), or otherwise intrinsically motivating.

Another key challenge is handling data sparsity. Although recent advances in large-scale pretraining have led to few-shot capabilities on many language tasks, interactive and grounded tasks pose additional challenges in sparsity. First, the addition of multimodal context widens the conditioning space. Second, many grounded settings involve domain-specific knowledge; for example, instruction following and grounded dialogue settings often have unique action spaces, as well as domain-specific abstractions or conventions in language use. Finally, many settings require adaptation to individual users, for which there will always be limited data.

As NLP expands to an ever-wider range of contexts, we encourage work to include pragmatics as a central component, with the goal of communicating successfully, efficiently, and naturally with people in challenging and useful settings.

## Limitations

Although we aim to describe a representative sample of tasks in Table 1, our coverage is necessarily incomplete, especially in domains such as image captioning, instruction-following, and collaborative dialogue, so we refer readers to other surveys on these issues (e.g., Luketina et al., 2019). As noted in Section 3, we focus exclusively on task-oriented grounded domains involving natural language data. Our survey therefore includes limited discussion of pragmatic phenomena in unimodal text domains such as chitchat dialogue, purely textual task-oriented dialogue, and language classification tasks (although c.f. Section 4.3 and Appendix B), and omits much work on analyzing the abilities of models to perform classic pragmatic tasks such as implicature and presupposition (e.g., Ross and Pavlick, 2019; Jeretic et al., 2020). We

also do not discuss tasks involving synthetic or emergent language, but see Lazaridou and Baroni (2020) for a survey of the latter.

While we focus mostly on written or typed language, there is also some computational work that has focused on spoken language pragmatics in grounded settings (Harwath et al., 2016; Sharma et al., 2018) as well as work in linguistics at the intersection of speech and pragmatics, e.g., focused on prosody (Pierrehumbert and Hirschberg, 1990; Sedivy et al., 1999).

Our discussion of modeling frameworks for pragmatics in Section 4 focuses on approaches that distinguish between semantics and pragmatics through social reasoning about other agents' beliefs and goals. Due to space limitations, we did not discuss alternate theories proposing that pragmatically enriched meanings are derived within the grammar of a language, without recourse to probabilistic social reasoning (e.g., Fox, 2007; Chierchia et al., 2012; Asherov et al., 2021). These theories remain difficult to implement at scale, but we encourage future work to explore them as candidate hypotheses alongside the frameworks discussed in Section 4.

There is also rich body of work on formalizing and modeling discourse context beyond the approaches we cover here, including conversational analysis (Schegloff, 1968; Sacks et al., 1974) and discourse coherence and structure (Hobbs, 1979; Grosz and Sidner, 1986; Webber, 1991; Kamp and Reyle, 1993; Grosz et al., 1995; Webber et al., 2003; Asher and Lascarides, 2003; Barzilay and Lapata, 2008). We refer to Cohen et al. (1990), Clark (1996), Jurafsky and Martin (2014), and Alikhani and Stone (2020) for entry points.

## Acknowledgments

We are grateful to Alane Suhr, Justin Chiu, Jessy Lin, Kevin Yang, Ge Gao, Ana Smith, and Herbert Clark for early discussions that led to this survey. We also thank Chris Potts, Alane Suhr, Ari Holtzman, Victor Zhong, Laura Rimell, Chris Dyer, Saujas Vaduguru, Tao Yu, Allen Nie, Dan Klein, Andrew Lampinen, and numerous anonymous reviewers and ACs for their comments on earlier drafts of our paper. Nicholas Tomlin was supported by the DARPA XAI and LwLL programs and a NSF Graduate Research Fellowship. Jennifer Hu was supported by a NSF Graduate Research Fellowship and a NSF Doctoral Dissertation Research Improvement Grant.

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

# A Types of Context

In this section, we outline the broad types of context that lead to pragmatic enrichment of language, and point readers to Levinson (2000), Birner (2012), or Yule (1996) for a more comprehensive discussion. In this paper we focus mainly on visual and embodied contexts, for several reasons. First, human communication is typically situated in settings with modalities beyond language, which makes it important to capture in order to build NLP models that interact naturally with humans in the world. Indeed, recent work has argued that grounding is an essential component of language understanding (e.g., Bisk et al., 2020; Bender and Koller, 2020). Second, visual and embodied settings introduce enough complexity to elicit interesting linguistic behaviors and serve as a challenge for models, while still allowing researchers to control experimental aspects of the tasks. Finally, there has been a rapid increase in research on multimodal language learning, which makes studying pragmatics in these models and tasks timely and relevant.

## A.1 Common Ground

To communicate successfully, speakers and listeners need to maintain a shared set of information, taken collectively to be *common ground* (e.g., Lewis, 1969; Stalnaker, 1978; Clark and Brennan, 1991; Traum, 1994; Stalnaker, 2002; Clark, 2015). A large body of work has demonstrated that humans produce and comprehend language in ways that depend on assumptions about the knowledge of their communicative partners (e.g., Krauss and Weinheimer, 1966; Horton and Keysar, 1996; Nadig and Sedivy, 2002; Clark and Bernicot, 2008; Hilliard and Cook, 2016; Yoon and Brown-Schmidt, 2019; Hawkins et al., 2021). Even in one-shot encounters where there is minimal partner-specific knowledge, the success of computational models of pragmatics (Frank and Goodman, 2012; Goodman and Frank, 2016) suggests that humans leverage a rich set of shared assumptions in pragmatic communication – from broad expectations that their partners abide by cooperative principles (e.g., Grice, 1975; Horn, 1984) to fine-grained knowledge of the potential utterances, meanings, and utterance-meaning mappings under joint consideration. Below, we discuss some key elements of common ground that give rise to pragmatically enriched meanings in naturalistic communication.

**Norms of Interaction.** As language is a social behavior, speakers and listeners typically abide by a set of norms. For example, Grice (1975) argues that it is generally understood that conversational partners act cooperatively and rationally. Grice also proposes a set of maxims that govern communication—rational speakers should be concise, informative, and relevant. These norms in turn give rise to a variety of nonliteral inferences known as *conversational implicatures*. Suppose, for example, Alice says to Bob: "Carl ate some of the cookies that we baked for the party". Bob likely draws the inference that Carl did not eat *all* of the cookies, even though the literal meaning of the utterance – that Carl ate at least one of the cookies – is logically compatible with such a scenario. This inference can be explained in the following way: if Alice knows that Carl ate all the cookies, and if she wants to be informative, then she would have said "Carl ate all of the cookies" instead.

**Goals and Joint Actions.** In addition to general norms of interaction, the particular social or task-related goals that elicit a linguistic expression can affect its meaning. The theory of *speech acts* (Searle, 1969; Austin, 1975) frames utterances (e.g., "please stand up") as actions on several levels: *locutionary*, the utterance itself; *illocutionary*, the intention (e.g., asking the listener to stand up); and *perlocutionary*, the actual effect that the action has in the world (e.g., the listener stands up). Context can have strong effects on the illocutionary and perlocutionary levels. This is particularly true for formal speech acts which can only take effect under *felicity conditions*, e.g. making a promise, or performing a marriage, but also occurs in commonplace situations e.g., asking "Did you get my email?" might be an indirect request to reply, or a direct question while debugging an internet connection. More generally, interlocutors typically recognize that they are undertaking *joint activities* together with their partners (Clark, 1996) and try to collaboratively plan and act to coordinate on and realize the relevant goals. These shared goals provide a source of context that enriches language.

**Common Knowledge.** Interpretation is aided by prior information that interlocutors bring to an interaction. For example, suppose Alice asks "What color was the woman's scarf?" and Bob answers "green". If Bob is a fashion designer with a keen eye for color palettes, this might implicate that

the scarf was a rather prototypical shade of green, and not olive green or chartreuse. On the other hand, if Bob doesn't know many specific color terms, Alice doesn't have grounds to infer that Bob meant to refer to a specific subspace of green. The world knowledge and commonsense relationships shared by conversational partners can also give rise to scalar implicatures formed by ad-hoc ordering relationships (Hirschberg, 1985b) and lead to pedagogic behavior (Chai et al., 2019).

**Discourse Context.** Communication is most often not a one-shot utterance, but instead unfolds over time. As a document or a conversation proceeds, the common ground can be updated with new information from the discourse context. At a basic level, discourse context includes previously-established information which can be referred to later on, whether explicitly (e.g., *a dog bounded into the room... it barked*) or implicitly (e.g., *a dog bounded into the room... Sam was surprised*). Information can also be introduced implicitly, for example through presupposition and accommodation (e.g., *Alex stopped smoking* presupposes that Alex smoked). Implicitly-introduced information can in some cases (implicature) also be reinforced or denied, e.g., *Carl ate some of the cookies; indeed, he ate all of them!*.

### A.2 Multimodal Context

So far, we have discussed aspects of context given by social or linguistic factors. While all of the above types of context also arise in grounded and multimodal settings, the physical context in which communication is situated also plays an additional component in deriving linguistic meaning. As mentioned above, we focus on visual and embodied contexts in this paper, as these contexts reflect naturalistic communication while also allowing for fine-grained experimental control.

**Visual.** Visual context serves to disambiguate and enrich the language of meaning on multiple levels. On a level close to semantics, visual context can disambiguate word senses: e.g., "bank" likely has a different meaning in the caption of a photo of a river than in a photo of a city street. Referring expressions (e.g., *the red one*) often can only be resolved in a visual context, and deictic expressions, like English *here, there, this* and *that*, are frequently used in language to individuate referents in their immediate context, relying on mutual knowledge of what the speaker and listener can see (Clark

and Marshall, 1981). Reference intepretation can also be affected by the location of the speaker and hearer in the world (Birner, 2012), and can involve physical analogues of implicature (e.g., *the black one* might be a good description for a dark grey object if all other visible objects are lighter) (Golland et al., 2010; Udagawa et al., 2020).

**Embodied.** Facial expressions, gaze, and gestures (Cassell et al., 1994; Traum and Rickel, 2002; Sidner et al., 2005; Prasov and Chai, 2008; Bohus and Horvitz, 2010; Koller et al., 2012; Yu et al., 2015) can aid interpretation if they are available, e.g., a speaker first making eye contact with a listener, then looking at an intended object. Speakers can issue corrections if they are able to observe a listener carrying out actions (Clark and Krych, 2004; Koller et al., 2010; Thomason et al., 2019; Suhr et al., 2019a), and the physical movements of the listener can intentionally convey uncertainty (Hough and Schlangen, 2017) and intent (Dragan et al., 2013). Physical properties of the environment and tasks (Chai et al., 2019) and the capabilities of the speaker and listener (Chai et al., 2014), also affect the interpretation and generation of commands and requests — e.g., the classic pragmatic example *Can you pass the salt?*, which typically is an indirect request when spoken to a person, may have a literal interpretation when spoken to a robot with a faulty gripper.

## B Unimodal Pragmatics

Although we primarily focus on the role of pragmatics in grounded environments, several text-only tasks that emphasize specific pragmatic phenomena also exist. For example, IMPPRES (Jeretic et al., 2020) and NOPE (Parrish et al., 2021) are benchmark datasets designed to test whether large language models can reliably predict implicatures and presuppositions, respectively. Similarly, Schuster et al. (2020) and Li et al. (2021) evaluate the ability of sentence encoding models to predict the rate at which humans draw scalar implicatures. Other datasets like the Self-Annotated Reddit Corpus (SARC) for sarcasm detection (Khodak et al., 2018) may also be viewed as pragmatic in nature (Kolchinski and Potts, 2018). While these datasets are limited to unimodal text, they have two main advantages over many multimodal tasks: (1) many unimodal pragmatic datasets are naturally-occurring, resulting in larger datasets with more realistic language, and (2) all of these datasets fo-

cus on specific pragmatic phenomena, such as pre-supposition. We suggest that future work on multimodal pragmatics should take inspiration from these properties and build larger and more targeted datasets.

A separate body of work has investigated situated language understanding through interactive fiction (IF) games (e.g., Ammanabrolu and Riedl, 2021; Hausknecht et al., 2020; Urbanek et al., 2019). IF games offer a framework for investigating goal-driven linguistic behaviors in a dynamic, richly structured world. Players observe natural-language descriptions of the simulated world, take actions via natural language, and receive scores based on their actions. The simulations are also partially observable, in that players must reason about the unerlying world state through incomplete textual descriptions of immediate surroundings. In this way, IF games avoid some of the practical issues of grounding in visual environments, while still requiring actions to be situated in rich, dynamic contexts. Furthermore, Shridhar et al. (2021) demonstrate that commonsense priors learned through IF games can be leveraged for better generalization in visually grounded environments, suggesting that text-only games induce representations that can be adapted to multimodal settings.