# OpenReview forum: "Pragmatics in Language Grounding: Phenomena, Tasks, and Modeling Approaches"
_EMNLP/2023/Conference — EMNLP 2023 Findings_

### Official Review · Reviewer_3CNV · 2023-07-25

**Soundness:** 4

**Excitement:**

5: Transformative: This paper is likely to change its subfield or computational linguistics broadly. It should be considered for a best paper award. This paper changes the current understanding of some phenomenon, shows a widely held practice to be erroneous in someway, enables a promising direction of research for a (broad or narrow) topic, or creates an exciting new technique.

**Missing References:**

The authors were thorough, but a very recent paper that I came across I think is relevant here. Though the author doesn't mention pragmatics, there are some similar arguments that I think are relevant to the discussion:

https://arxiv.org/abs/2307.04518

**Paper Topic And Main Contributions:**

This paper reviews how pragmatics are under-studied, yet important in how they affect language grounding. Though not the typical paper that introduces a new model or dataset, this paper is quite comprehensive in its depth and breadth of the relevant literature.

**Questions For The Authors:**

- Suggestion: be careful about the word "context" because in most NLP papers it means textual/lexical context, but the authors mean physical or world context. Please be clear.
- The paper focuses on pragmatics, but some could argue that there is a lot of overlap with semantics. Can the authors delineate further the semantics-pragmatics interface and how they related to each other?
- There is a bit of an abrupt jump from the end of section 2 to the start of section 3. Please make the transition smoother.
- I think it would be helpful to mention the claim that LLMs have emergent behaviors, which may include pragmatic qualities. Can pragmatics follow from textual learning?

**Reasons To Accept:**

- The topic is beyond timely. Taking a long, hard look at pragmatics is long overdue.
- The paper is well-written, easy to follow with plenty of references and citations to back up any claims.
- The focus is actually on an important aspect of language, not just on models.

**Reasons To Reject:**

- There's no new proposed model or evaluation, which is traditional at EMLNP, but I don't think that should be a reason to reject.

**Reproducibility:**

N/A: Doesn't apply, since the paper does not include empirical results.

**Reviewer Confidence:**

5: Positive that my evaluation is correct. I read the paper very carefully and I am very familiar with related work.

---

> ### Author Rebuttal · Authors · 2023-08-28
>
> Thank you for your review! We’re glad that you found our paper well-written and important and appreciated our focus on linguistic phenomena. In response to specific comments:
>
> > be careful about the word "context"
>
> This is a good point. We provide some discussion of types of context in Sec 2.1 / Appendix A but will aim to be more precise about the type of context we refer to (typically physical or world context)throughout the paper.
>
> > Can the authors delineate further the semantics-pragmatics interface and how they related to each other?
>
> The boundary between semantics and pragmatics was also mentioned by tVhy. We agree that the boundary between semantics and pragmatics is quite grey, and if we had to take a stance, our position largely holds with Birner 2012: semantics largely concerns truth-conditional, or context-independent, meaning of an utterance; while pragmatics largely concerns context-dependent meaning, beyond truth-conditional. The truth-conditional & context-independent definitions of the boundary do not produce the same distinction on all phenomena (e.g. anaphoric pronouns; conventional implicature), but as we are most interested in this paper in grounded phenomena which are both heavily-contextual and deal with non truth-conditional meaning, we chose to not delineate the boundary, as it seemed less relevant to our intended audience.
>
> > Can pragmatics follow from textual learning?
>
> This was also mentioned by reviewer tVhy. Our response: “We agree with the reviewer that this is a very important, and open question, and we do mention it in our discussion/future work section (lines 660-665). However, since our work is primarily a survey paper, and there’s not (yet) a good answer to this question in the literature, we don’t spend much time on it. We will emphasize this question, and that we view it as still open, in any future version of the paper."
>
> Thank you also for the suggested reference! It looks very interesting and relevant.

---

### Official Review · Reviewer_tVhy · 2023-08-04

**Soundness:** 3

**Excitement:**

4: Strong: This paper deepens the understanding of some phenomenon or lowers the barriers to an existing research direction.

**Missing References:**

- It seems to me that the ReferItGame papers would fit perfectly in this taxonomy, as the setting brilliantly reduces interaction to its minimal form (but not beyond), in that the speaker knows that a listener is present. It also predates much of the work that is cited.
Kazemzadeh, S., Ordonez, V., Matten, M., & Berg, T. L. (2014). ReferItGame : Referring to Objects in Photographs of Natural Scenes. In EMNLP 2014 (pp. 787–798).

- For a more comprehensive argument about the role of interaction (rather than just "pragmatics"), could cite:
Dingemanse, M., Liesenfeld, A., Rasenberg, M., Albert, S., Ameka, F.K., Birhane, A., Bolis, D., Cassell, J., Clift, R., Cuffari, E., De Jaegher, H., Novaes, C.D., Enfield, N.J., Fusaroli, R., Gregoromichelaki, E., Hutchins, E., Konvalinka, I., Milton, D., Rączaszek-Leonardi, J., Reddy, V., Rossano, F., Schlangen, D., Seibt, J., Stokoe, E., Suchman, L., Vesper, C., Wheatley, T. and Wiltschko, M. (2023), Beyond Single-Mindedness: A Figure-Ground Reversal for the Cognitive Sciences. Cognitive Science, 47: e13230. https://doi.org/10.1111/cogs.13230


**Paper Topic And Main Contributions:**

This position paper argues that NLP has so far not looked closely enough at language understanding tasks where there is a rich context to which the used language implicitly makes reference. The paper goes into some types of such contexts, and discusses tasks that the authors argue would bring them out and make them available for current methods.

The paper presents a survey of foundational work that could especially be useful for researchers coming into this field from outside of linguistics (where of course the role of pragmatics has been known at least since Peirce, Bühler, and Morris).

The motivation given in the introduction rests at least rhetorically a bit on an outdated conception of the division of labour between semantics and pragmatics -- in the claim that existing benchmarks measure literal meaning (line 042), which of course QnA datasets, or any dataset with material that goes beyond a single sentece, do not do, as pragmatic enrichment phenomena are present even in purely linguistic contexts -- but that is a minor point.

Also, the paper does not address (or even just mention) the question -- not at all obvious to this reviewer -- of whether existing methods based on approximating functions from data of example function applications simply "scale up" to interactive settings.

However, overall, I think this paper will be very valuable to the EMNLP community, and I recommend acceptance.


**Questions For The Authors:**

- Would you say that the difference between typed language and spoken language is irrelevant? Or is the focus on typed language tasks due to methodological considerations?


**Reasons To Accept:**

- brings a familiar argument from linguistics and psycholinguistis (language primarily is an instrument to be used, in particular contexts and particular settings) to the attention of NLP researchers, who tend to overlook this

- provides a useful systematization of tasks / datasets, connection features of the task to types of relevant context

- provides useful guidance for the design of challenging next tasks


**Reasons To Reject:**

- does not mention the active research fields that look at situated interaction, outside of narrowly defined NLP (e.g., in HRI, in conversational analysis); only mentions the classics

- underplays the role of interaction in creating the kind of dynamic contexts that mark situated language use; makes no argument for the implicit claim that there is no qualitative difference between functional *tasks* like captioning and (single shot) instruction following on the one hand and interactive settings like "grounded goal-oriented dialogue" on the other


**Reproducibility:**

N/A: Doesn't apply, since the paper does not include empirical results.

**Reviewer Confidence:**

5: Positive that my evaluation is correct. I read the paper very carefully and I am very familiar with related work.

---

> ### Author Rebuttal · Authors · 2023-08-28
>
> Thanks for the thorough review! We’re glad that you found the paper valuable to the EMNLP community and appreciated our taxonomy of tasks and datasets, as well as our recommendations for future task design. In response to specific comments:
>
> > existing benchmarks measure literal meaning
>
> We agree that we presented a simplified distinction between semantics and pragmatics, and that unimodal tasks like question answering can leverage rich pragmatic phenomena. In practice, however, we suggest that many datasets for unimodal tasks like natural language entailment or factoid QA focus primarily on semantic understanding, rather than pragmatics.
>
> > whether existing methods…simply "scale up" to interactive settings
>
> We agree with the reviewer that this is a very important, and open question, and we do  mention it in our discussion/future work section (lines 660-665). However, since our work is primarily a survey paper, and there’s not (yet) a good answer to this question in the literature, we don’t spend much time on it. We will emphasize this question, and that we view it as still open, in any future version of the paper.
>
> > does not mention the active research fields that look at situated interaction
>
> Thank you for pointing this out. While we did not include references to conversational analysis in the main body due to space restrictions, we did mention them in the limitations section (lines 758-768), and we cite some work from HRI briefly in the main body (Dragan et al. 2013, line 1945). We are happy to move some of this supplementary to the main text / expand the citations and descriptions given more space in a future version.
>
> >  implicit claim that there is no qualitative difference between functional tasks and interactive settings
>
> We didn’t intend to make such a claim, implicitly or explicitly. In contrast, we argue that the richer, interaction-heavy domains like grounded dialogue often elaborate a richer set of pragmatic phenomena (e.g., lines 603-604). We are happy to make this more explicit in the paper, or clarify any particular unclear parts that might have unintentionally made this claim.
>
> > Would you say that the difference between typed language and spoken language is irrelevant? Or is the focus on typed language tasks due to methodological considerations?
>
> We wouldn’t consider the distinction irrelevant, but we are aware of relatively little computational work that has focused on spoken language pragmatics in grounded settings, with a few exceptions being Harwath et al. 2016 [1] and Sharma et al. 2018. There also some great linguistics work at the intersection of speech and pragmatics, though, e.g., focused on prosody (Pierrehumbert & Hirschberg 1990 [3]) and incrementality (Sedivy et al. 1999 [4]).
>
> [1] Harwath et al. Unsupervised Learning of Spoken Language with Visual Context. NeurIPS 2016
>
> [2] Sharma et al. Conceptual Captions: A Cleaned, Hypernymed, Image Alt-text Dataset For Automatic Image Captioning. ACL 2018
>
> [3] Pierrehumbert & Hirschberg. The Meaning of Intonational Contours in the Interpretation of Discourse. Intentions in Communication, 1990
>
> [4] Sedivy et al. Achieving incremental semantic interpretation through contextual representation. Cognition, 1999

---

### Official Review · Reviewer_ED4y · 2023-08-04

**Soundness:** 1

**Excitement:**

2: Mediocre: This paper makes marginal contributions (vs non-contemporaneous work), so I would rather not see it in the conference.

**Paper Topic And Main Contributions:**

The paper aims to remind the NLP community (which today is heavily focusing on machine learning) of the fact that language understanding is strongly related to language use in situated interactive scenarios. The authors provide an overview of pragmatics phenomena and list a range of task settings where different aspects of grounded dialogue occur. The authors argue that LLMs should be studied in the light of their (lack of) capabilities to handle pragmatic phenomena.

**Reasons To Accept:**

The paper provides a useful overview of pragmatics phenomena and relevant task settings combined with a comprehensive reference list. The authors thematize the lack of pragmatics in current LLMs, and, in my opinion, rightly suggest to study and further develop (multimodal) LLMs in tandem with pragmatics modelling.

**Reasons To Reject:**

The paper makes a strong claim to take pragmatics serious when developing (multimodal) LLMs. It gives a good account of pragmatics, however, remains vague on how pragmatically informed tasks should be utilized to further develop LLMs. This imbalance it reflected in the room given to the topic of pragmatics  (~ 7 pages) versus the considerations related to pragmatic modelling and LLms (less than one column).

**Reproducibility:**

N/A: Doesn't apply, since the paper does not include empirical results.

**Reviewer Confidence:**

5: Positive that my evaluation is correct. I read the paper very carefully and I am very familiar with related work.

---

> ### Author Rebuttal · Authors · 2023-08-28
>
> Thank you for your review! We’re encouraged that you appreciated our account of pragmatic phenomena and tasks and found the cited work to be comprehensive.
>
> The primary goal of our paper is to provide an account of pragmatics, targeted toward an NLP audience. As a result, the bulk of our paper does focus on pragmatics. However, while a linguistically-oriented introduction to pragmatics would typically focus on specific types of pragmatic phenomena (e.g., implicature, presupposition, deixis), our work emphasizes the functional roles of pragmatics (sec 2.2) and how they relate to grounded language tasks. As noted in your summary, most of the NLP community is currently focused on machine learning, so we hope this alternative perspective will be useful to practitioners thinking about developing NLP systems, especially in grounded domains.
>
> We would like to note that, while we agree that our paper does spend most of its length (7 pages) talking about pragmatics, much of this discussion is also highly compatible with LLMs. LLMs have already begun to be put to use to tackle the grounded pragmatic tasks we outline in Table 1 and Section 3. For example, Kojima et al. 2021 fine-tunes a pre-trained GPT-2 model to adapt to people for instruction generation in CerealBar [1]. The pragmatic methods in Section 4 are also quite compatible with LLMs, e.g. Liu et al. 2023 combine RSA with meta-learning to apply GPT models in an image reference game setting [2]; Meta et al. 2022 use a large BART model in conjunction with a multi-agent planning procedure in the grounded dialogue game of Diplomacy [3]. As vision-to-LLM adapters (e.g. LIMBER, Flamingo, Fromage, MAGMA [4-7]) continue to improve, we expect to see more work applying LLMs as components of pragmatic models for these grounding pragmatic tasks. Our discussion section (Section 5) does touch on this LLM+pragmatics synthesis, and we will use the extra page in any future version of the paper to expand this discussion.
>
> We also provide some discussion of datasets for evaluating unimodal pragmatics (primarily in LLMs) in Appendix B.
>
> [1] Kojima et al. Continual Learning for Grounded Instruction Generation by Observing Human Following Behavior. TACL 2021
>
> [2] Liu et al. Computational Language Acquisition with Theory of Mind. ICLR 2023
>
> [3] Meta et al. Human-Level Play in the Game of Diplomacy by Combining Language Models with Strategic Reasoning. Science 2022
>
> [4] Merullo et al. Linearly Mapping from Image to Text Space. ICLR 2023
>
> [5] Alayrac et al. Flamingo: a Visual Language Model for Few-Shot Learning. NeurIPS 2022
>
> [6] Koh et al. Grounding Language Models to Images for Multimodal Inputs and Outputs. ICML 2023
>
> [7] Eichenberg et al. MAGMA -- Multimodal Augmentation of Generative Models through Adapter-based Finetuning. EMNLP 2022

---

### Meta-Review · Area_Chair_PCqj · 2023-09-15

**Recommendation:** 4

**Metareview:**

Reviewers tVhy and 3CNV provide a positive assessment of validity and are quite excited about the paper, whereas Reviewer ED4y gives low scores on both dimensions.

Synthesizing the reviews, the following major strengths or weaknesses were mentioned:

Strengths

- provides a useful overview of pragmatics (R1, R3) for a community that may overlook relevant key properties of language (R2, R3), provinces guidance for relevant challenging tasks (R2)

Weaknesses

- remains vague on how pragmatics could actually come to play in developing LLMs (R1)
perceived relatively narrow focus in the way linguistic interaction is framed and conceived (R2)

Reconciling the disparate scores given by reviewers:
The weakness mentioned by R1 (ED4y) – i.e., remaining vague on how pragmatics can come to play in developing LLMs – appears to be largely related to excitement rather than soundness, as it does not undermine the validity of the conclusions drawn by the paper or the arguments made in it. Thus, I believe their low soundness score (1) is not well justified by the review and can be discounted.
On the other hand, the three excitement scores (2,4,5), while variable, are supported as subjective assessments by the three reviewers.

---

### Decision · Program_Chairs · 2023-10-07

**Decision:**

Accept-Findings

**Comment:**

Reviewers tVhy and 3CNV provide a positive assessment of validity and are quite excited about the paper, whereas Reviewer ED4y gives low scores on both dimensions.

Synthesizing the reviews, the following major strengths or weaknesses were mentioned:

Strengths

- provides a useful overview of pragmatics (R1, R3) for a community that may overlook relevant key properties of language (R2, R3), provinces guidance for relevant challenging tasks (R2)

Weaknesses

- remains vague on how pragmatics could actually come to play in developing LLMs (R1)
perceived relatively narrow focus in the way linguistic interaction is framed and conceived (R2)

Reconciling the disparate scores given by reviewers:
The weakness mentioned by R1 (ED4y) – i.e., remaining vague on how pragmatics can come to play in developing LLMs – appears to be largely related to excitement rather than soundness, as it does not undermine the validity of the conclusions drawn by the paper or the arguments made in it. Thus, I believe their low soundness score (1) is not well justified by the review and can be discounted.
On the other hand, the three excitement scores (2,4,5), while variable, are supported as subjective assessments by the three reviewers.